# Collagen-Based Materials Modified by Phenolic Acids—A Review

**DOI:** 10.3390/ma13163641

**Published:** 2020-08-17

**Authors:** Beata Kaczmarek, Olha Mazur

**Affiliations:** Department of Biomaterials and Cosmetics Chemistry, Faculty of Chemistry, Nicolaus Copernicus University in Toruń, Gagarin 7, 87-100 Toruń, Poland; 289185@stud.umk.pl

**Keywords:** collagen, tannic acid, ferulic acid, caffeic acid, gallic acid

## Abstract

Collagen-based biomaterials constitute one of the most widely studied types of materials for biomedical applications. Low thermal and mechanical parameters are the main disadvantages of such structures. Moreover, they present low stability in the case of degradation by collagenase. To improve the properties of collagen-based materials, different types of cross-linkers have been researched. In recent years, phenolic acids have been studied as collagen modifiers. Mainly, tannic acid has been tested for collagen modification as it interacts with a polymeric chain by strong hydrogen bonds. When compared to pure collagen, such complexes show both antimicrobial activity and improved physicochemical properties. Less research reporting on other phenolic acids has been published. This review is a summary of the present knowledge about phenolic acids (e.g., tannic, ferulic, gallic, and caffeic acid) application as collagen cross-linkers. The studies concerning collagen-based materials with phenolic acids are summarized and discussed.

## 1. Introduction

Regenerative medicine gives almost magical possibilities, but fortunately, this fact-based field has little in common with magic. It is the common denominator for many medical specialties—from orthopedics or ophthalmology to oncology, neurology, dermatology, or urology; it is based on specialist knowledge in various fields, e.g., biomedical engineering, regenerative surgery, and biotechnology. It is also an extremely prospective branch of activities of scientific centers specializing in the use of stem cells or biomedical engineering. Above all, it gives great hope to patients who lack therapeutic solutions in the available range of currently used treatments or therapies [1].

In general, the main purpose of regenerative medicine is to repair, restore, and regenerate tissues and organs that have been damaged as a result of illness, congenital defect, or injury, but also the aging process of human or animal bodies, because it is also used in veterinary medicine therapies.

Regenerative medicine includes cell therapy, tissue, or biomedical engineering products and, in some cases, gene therapies. The formal classification of products depends on the registration status or the authorization procedure. Numerous advanced medicinal products offered are inspired by regenerative medicine achievements [2].

The first bone marrow transplantation in leukemia performed in 1956 is considered the beginning of regenerative medicine. Therefore, only within the last two or three decades, the current state of this discipline has been shaped, and it seems that the greatest discoveries and ultramodern technologies are yet to come. This is a huge research field for specialist scientific centers, so monitoring the progress is a necessity. In the following years, we can expect significant advances in regenerative medicine. We are implementing projects inspired by science fiction novels, such as complete regeneration of a lost limb or the growth of a new organ and its auto-implantation in a disease or an injury site (liver, heart, or kidney). Research is being carried out on each organ, not only to completely replace but also to partially regenerate damaged tissues [3].

Stem cells are the main “ingredient” used in regenerative medicine. It is a pool of non-specialized cells with the potential to transform into cells of any tissue. Their task in the body is to replace “old” cells with new ones, which is why they have the unlimited ability to divide and differentiate. Depending on the source of acquisition, two types are distinguished: embryonic and, with lower potential, stem cells of already developed organisms [4]. Stem cells taken from a donor showing tissue compatibility with the recipient (allogeneic transplant) or the patient (autologous transplant) may be used here. In the second case, the problematic deficit of suitable donors, the risk of rejection, or complications resulting from prolonged intake of immunosuppressants preventing recoil are avoided [5].

In regenerative medicine solutions, its interdisciplinary character seems to be the key issue. To create a new product, having complete knowledge in one specialty is not enough, and the cooperation of specialized centers from various fields is essential. To fully utilize the potential of stem cells, necessary conditions must be met to support the creation of new tissues. Such elements include growth factors, stimulating differentiation, and the “skeleton” on which the new tissue superstructure will be organized along with the network of blood vessels.

The source of mesenchymal stem cells is not only bone marrow but also peripheral blood and other tissues. For example, adipose tissue or synovium of the joints are sources of stem cells already used in specialized orthopedic centers or sports medicine for osteoarthritis treatment, reconstruction of articular cartilage, treatment of muscle damage, tendon attachments, and meniscus regeneration.

The concept of regenerative medicine is so broad that it can also involve products authorized as medical devices. For example, platelet-rich plasma is widely used in orthopedics, but also in dentistry or aesthetic medicine [6,7]. Separation kits for such plasma require standard notification as a medical device. We are not dealing here with complete tissue reconstruction, but the effect on regeneration and renewal seems to be indisputable.

Regenerative medicine will be a driving force for scientists in many fields and will surprise us in the coming years more than once, and its interdisciplinarity seems to be most important. Because in this case unlimited possibilities are opening for centers with various specialties, ultramodern technologies combine many fields.

Regenerative medicine is one of the fastest-growing contemporary issues. It is estimated that the global regenerative medicine market was worth over 290 billion USD in 2015. This interdisciplinary field is aimed at exchanging, creating, or regenerating human cells, tissues, or organs in order to restore the normal functioning of the body. It mainly meets the needs of patients suffering from diseases resistant to conventional treatment methods requiring specific therapies or transplants, but it also creates a great opportunity for more effective treatment and shorter recovery time for other groups of patients [8].

Scientists are looking for biomaterials with properties imitating natural ones, the properties of which are as close as possible to the features of natural tissues in the human body. The purpose of this research is to create a material accelerating wound healing that does not create an immune response from the patient. The ideal solution would be to create a biomaterial with properties similar to those of the organ it is intended to replace or support [9]. Conventional materials for dentures used for many years are metals. They are used to fill in, among others, bone defects in complicated limb fractures or spine surgery. The materials used are titanium alloys, which have been a standard in implantology for a long time. They have good mechanical properties, but their use is associated with the risk of implant rejection. Above all, however, these materials will never replace lost tissues or be fully biocompatible with the patient’s body. That is why in recent years, research on the use of polymers as materials for the production of implants and prostheses has begun. As a consequence, an increasingly growing number of polymer materials have been applied in surgery, medicine, and transplantation. These biomaterials can be made of pure polymers and composites, i.e., mixtures of several polymers with the addition of metals (i.e., silver, gold, zinc, chromium, copper, cobalt, titanium), ceramic materials, etc. [10].

Polymers are widely used as basic compounds for advanced materials production, and they are found in almost every material used in our daily life. The importance of polymers has been even more emphasized because of their applications in different branches of science, technology, and industry—from basic uses to biopolymers and therapeutic polymer production. It is not surprising that nearly all material scientists and more than half of all chemists and chemical engineers, a large number of physicists, textile technologists, mechanical engineers, pharmacists, and other groups of scientists are involved in research and development projects related to polymers [11].

Natural polymers are defined as materials that widely occur in nature or are extracted from natural, both plant and animal, sources. The challenge of using natural polymers or developing new materials for producing goods used every day is connected with not only understanding their mode of action but also correct coordinating of the complex interplay between chemistry, biology, physics, and engineering. However, the main challenges scientists are currently facing are reducing manufacturing costs and scaling up their modifications, which will allow using those polymers in different environments to provide new solutions in the material industry.

Materials based on natural polymers present many advantages in their use to manufacture usable products such as biomaterials, packaging materials, cosmetics, etc. However, they are characterized by low stability in wet conditions, which is the main limitation of their potential application. Biopolymers can be modified in the cross-linking process, which involves combining external physical and chemical factors to modify biopolymer properties [12].

Due to the extensive use of natural polymers in industry, there is a need to search for methods that result in improving their properties. Biopolymer modification is one of the main interests of scientists in the polymeric field of science. Various methods have already been tested, such as the complexation by metals (e.g., Fe(III), Pb(II), Cu(II)), or the use of cross-linkers. Cross-linkers are compounds that may be added to polymeric solutions in which chemical reactions occur.

An effective modifier which could act as a biopolymer cross-linker showing antimicrobial properties needs to be found, as it may turn out to be a solution to various health care problems. An increase in antimicrobial resistance has a significant impact on public health, global development, and even the global economy. Furthermore, a constant increase in antimicrobial resistance is a reason for longer hospitalization time and expensive intensive care. Some biopolymers present antiviral activity which, considering the current situation, takes on a new significance [13].

There is a demand for an effective modifier which could act as a biopolymer cross-linker with antimicrobial properties, as it may turn out to be a solution to the above-mentioned health care problems. In order to enhance the properties of such materials, non-toxic additives are also required. In the past few years, an increased interest in polyphenolic acids has been observed [14].

From a biological science point of view, polyphenolic acids are essential due to a range of unique biological properties they possess [15]. They are involved in protecting human organisms against chronic degenerative diseases [16]. They are known to present preventive properties against many diseases, i.e., cardiovascular diseases, osteoporosis, neurodegenerative diseases, various forms of cancer, and diabetes mellitus [17,18]. Moreover, polyphenols demonstrate antioxidant properties against the cellular metabolic process and may inhibit cell propagation and apoptosis. Therefore, they have been researched as active additives for biopolymer-based materials.

Over the past few years, a growing interest in developing novel materials for biomedical materials has been observed. Since human organisms are exposed to different external factors which may cause serious diseases, an increased interest in naturally derived compounds has been observed, mainly due to the “from nature to nature” trend. Biopolymers—proteins and polysaccharides—have been employed in implantation and dental treatment, as metal coatings and wound dressing. Despite their excellent biocompatibility, the main issue is that the physicochemical properties of biopolymers are rather poor. Additionally, the above mentioned materials do not show activity against microbes. Thereby, they require modifications to improve both physicochemical and antimicrobial characteristics.

An effective modifier that could act as a biopolymer cross-linker with antimicrobial properties needs to be found, as it may turn out to be a solution to various health care problems. In order to enhance the properties of such materials, non-toxic additives are also required. Biopolymers interact with phenolic acids by strong hydrogen bonds and, as a result, complexes are formed. On the one hand, they possess antimicrobial properties, and on the other hand, the biopolymers’ physicochemical properties are improved. In the past few years, an increased interest in polyphenolic acids has been observed.

Polyphenolic acids are organic compounds containing a phenolic ring with carboxyl and hydroxyl groups. They are biosynthesized naturally by plants and marine organisms from which they are commonly extracted. Phenolic acids may function as cross-linkers for biopolymers owing to a variety of functional groups present in their structure. Thereby, they may improve the physicochemical properties of biopolymers.

From the biological science point of view, polyphenolic acids are essential due to a range of unique biological properties they reveal. They are involved in protecting human organisms against chronic degenerative diseases. They are known to present preventive properties against many diseases, i.e., cardiovascular diseases, osteoporosis, neurodegenerative diseases, various forms of cancer, and diabetes mellitus. Moreover, polyphenols demonstrate antioxidant properties against the cellular metabolic process and may inhibit cell propagation and apoptosis. Therefore, they have been researched as active additives for biopolymer-based materials.

Tannins, a special group of phenolic acids, are phenolic derivatives soluble in water whose molecular weight ranges between 500 and 3000 Da. The chemical structures of tannins depend on their origin, i.e., a plant species producing the compounds. The abovementioned acids occur naturally, mainly in complexes with polysaccharides and alkaloids.

Tannins may be isolated from natural sources and are safe for humans. However, each process aimed at isolating active compounds is complex. The content of tannins in a plant varies seasonally and depends on the part of a plant from which it is extracted (leaves, roots, etc.). Therefore, it is hard to obtain extracts with the same content of active compounds; the extraction problem requires significant scientific effort to be overcome.

This review focuses on the collagen-based material modified by phenolic acids. Collagen is the most common biopolymer used in regenerative medicine. It has been widely used as it presents high biocompatibility with tissues and is bioresorbable after implantation. However, collagen shows a high ability to degrade and has weak mechanical properties. In order to enhance its application possibilities, it has to be cross-linked.

## 2. Collagen

Collagen is a fibrous protein, or basically, a family of proteins responsible for providing strength and elasticity to animal and human tissues [19]. There are 29 types of collagen proteins, slightly different from each other in functional and structural terms [20]. Collagen is not present in plants, bacteria, and viruses. To meet the needs of biomedical, pharmaceutical, and cosmetic industries, it is obtained mainly from fungi, fish skin, and scales, as well as from rat tails, pig and beef tissues, and also from sea sponges, jellyfish, and egg capsules of the dogfish [21,22,23,24,25,26,27].

Collagen is insoluble in water and oils. As has been noticed, collagen extracted from tissues of young organisms is more susceptible to being soluble in hydrophilic solvents because its structure is equally or even more cross-linked than that observed in older organisms [28].

Additionally, the collagen solution is sensitive to UV radiation and high temperature values. Heat denaturation is strictly connected with breaking the bonds between the helixes, but the strong covalent bonds in the helixes remain intact. As a result, the formation of gelatin occurs. The main collagen drawback, however, is its poor thermal stability, which limits its industrial application. Denaturation temperature for fish collagen is about 33 °C and that for collagen isolated from rat tails is around 39 °C. Therefore, it is necessary to combine collagen with various additives, which increases its thermal stability [29]. However, the collagen of demosponge origin is thermostable up to 360 °C and needs no modification [30,31].

### 2.1. Types of Collagen

The most recognizable types of collagen are the following: type 1 collagen (skin, bones, tendons), type 2 collagen (cartilage), and type 3 collagen (vascularization). The type I collagen molecule is represented by a dextrorotatory helix consisting of three levorotatory alpha helices (tropocollagen). Each of the three chains is made up of about one thousand amino acids [32]. There are three amino acid residues per turn of the alpha-helix. Each alpha-helix contains glycine (which is always in the third place in the amino acid triads of collagen), proline, and hydroxyproline [33]. There are different types of bonds in the collagen molecule: hydrogen bonds, dipole–dipole and ionic interactions, covalent bonds, as well as van der Waals interactions [34]. However, intra-and intermolecular hydrogen bonds play an important role in collagen structure stabilization [29,35]. Type II collagen forms fibrils. Such a network allows cartilage tissue to provide tensile strength to the tissue. Such a type of collagen is easily degradable by proteolytic enzymes. Type III collagen is synthesized by cells as pre-procollagen. Three identical procollagen chains come together and form a helix that is stabilized by disulphide bonds.

### 2.2. Collagen Application

Due to its biological properties, non-toxicity, biocompatibility with all living organisms, and abundance, collagen is widely used as a biomaterial in pharmacy and medicine [36,37]. As collagen is characterized by high molecular weight, it does not have the ability to penetrate through the epidermis, works superficially on the skin limiting transepidermal water loss by creating a hydrophilic film, or exerting a protective effect by partially eliminating the action of anionic agents. Oral preparations containing collagen or collagen hydrolysates are used primarily for the prevention of diseases associated with changes in connective tissue (especially cartilaginous) and for cosmetic purposes to improve skin appearance. However, its effectiveness has been insufficiently studied. There are reports that orally applied preparations containing collagen peptides can improve the skin hydration level as well as reduce discomfort in the joints resulting from intense physical activity in young adults [27].

Collagen is used in aesthetic medicine as a filler to even out the skin surface in the areas where wrinkles or atrophic scars occur, and to model the mouth and the face oval shape. It was one of the first substances used for this purpose (right after fat). A distinction can be made between synthetic and natural fillers, and among the latter, xenogenic (from animal tissues, usually bovine), allogenic (from other tissues, deceased people), or autogenic (from own tissues) can be enumerated. Collagen preparations differ in the likelihood of causing allergies; some of them require allergy testing before use (in the case of bovine collagen, a double skin test) [38].

The use of collagen fillers can cause some side effects that can be classified into early and delayed reactions. The early ones include swelling, pain, itching, bruising, hard bumps formation, and infections. The delayed ones involve swelling, pain, nodules, scars and granuloma formation, and sterile abscesses. Currently, rather than collagen, hyaluronic acid derivatives have become more popular due to a lower risk of hypersensitivity they present, no necessity of prior skin testing, as well as a more prolonged effect. In the case of collagen, the filling effect usually lasts from three to a few more months [39].

### 2.3. Biomedical Aspect

Collagen and its derivatives are non-toxic and compatible with human tissues [40]. This is one of the reasons for their extensive use in the biomedical industry as tissue engineering and regenerative medicine material [41,42]. Collagen matrixes show low antigenicity and low inflammatory response. It has hemostatic properties and also promotes cell proliferation. However, due to its low thermal and water stability as well as weak mechanical properties, it has to be modified in the cross-linking process [43].

Collagen may be modified by physical methods, e.g., by exposure to UV irradiation, enzyme acting [44,45,46,47], or by dehydrothermal treatment and photooxidation [48,49,50,51,52,53,54,55,56,57,58,59]. Additionally, chemical cross-linkers may be added to collagen, including aldehydes (e.g., glutaraldehyde, formaldehyde, glyoxal) [60,61,62,63,64,65,66,67,68,69], genipin [70,71,72,73,74,75,76,77,78,79,80,81], polyphenols (e.g., phenolic acids) [81], a mixture of EDC/NHS [82,83,84], and starch dialdehyde [85,86,87,88,89,90,91]; they can also be cross-linked by mixing with other polymers, both natural (e.g., chitosan, hyaluronic acid) [92,93,94] and synthetic (e.g., polyurethane, polyvinyl alcohol) [95].

As a medical biomaterial, collagen in the form of atelocollagen (a type I collagen derivative with cleaved extensor peptides responsible mainly for protein immunogenicity) is usually used. It finds applications in regenerative medicine and tissue engineering. It is also employed in the production of implants and barrier membranes for tissue and bone regeneration, and in the closing of oral-sinus joints after tooth extraction. It is also used in drug delivery systems (DDS). Collagen biomaterials can be helpful, e.g., in the treatment of corneal damage and genitourinary system diseases.

Materials dedicated to biomedical applications should be safe when implanted into the human body. They cannot inhibit cell growth and activity. Thereby, collagen-based materials should be modified by natural compounds which may interact with collagen functional groups and, as a result, improve their properties. When discussing collagen modifiers, polyphenolic acids find wide interest as they are non-toxic. Moreover, they present antimicrobial and antioxidant activity. Therefore, their use can lead to enriching collagen-based materials with new features [96]. In order to modify collagen properties, phenolic acid addition has been examined. Different compounds such as tannic, ferulic, gallic, and caffeic acid have been tested.

### 2.4. Collagen–Tannic Acid

Tannic acid (TA; Figure 1) is a compound which belongs to the phenolic acid group. It has the most complex stereo structure with the highest number of hydroxylic groups [97]. The cross-linking process accompanying tannic acid addition to collagen does not affect the integrity of the triple helical structure of collagen [98]. Tannic acid has been widely studied as a collagen modifier, since it is able to interact with hydrophilic functional groups from a polymeric chain, both amino and carboxylic ones [99,100]. The comparison of collagen-based materials cross-linked by starch dialdehyde and tannic acid showed better biocompatibility in the case of materials with TA addition [101]. It may suggest that naturally derived compounds proposed as cross-linkers are more appropriate for materials dedicated to biomedical applications.

Moreover, tannic acid has antibacterial and antiviral properties; therefore, it is an interesting compound which may be used to fabricate bioactive materials for medical applications. Collagen forms a complex with tannic acid when solutions are mixed, because strong hydrogen bonds are formed. Such solutions may be prepared in acetate or citric buffers (pH = 4), which affects the physicochemical and antimicrobial properties of the final structures [102]. The study of tannic acid released from collagen–tannic acid complexes suggests that it has a killing effect for microbes which are not cytotoxic to human cells. The addition of tannic acid to collagen significantly decreases its hydrolysis rate and affects the antioxidant properties [103]. As tannic acid interacts with collagen by forming hydrogen bonds, it is released to the surrounding environment. The tannic acid release profile from Coll/TA matrixes follows the Peppas mechanism [98]. Materials based on collagen–tannic acid complexes showed better physical and chemical properties than those obtained from unmodified collagen [104]. The shrinking temperature was higher for such complexes, and they did not collapse under room conditions in an aqueous environment. Additionally, the hydrothermal stability of collagen modified by TA increased, and the degradation rate caused by collagenase decreased when compared to that for unmodified collagen [105].

Collagen has been cross-linked by tannic acid under microwave conditions, where the exposition to microwave activity gave thermal and hydrothermal stability higher than in the case of lack of such an exposition [106]. Microwaves improve the strength of hydrogen bonds between collagen and tannic acid and may be a useful tool for fabricating materials of stability higher than in the case of those manufactured in a traditional, simple manner, i.e., by mixing solutions under room conditions. Collagen-based materials are characterized by low thermal stability as their denaturation temperature depends on the collagen source. For collagen without TA, it is around 55 °C, and for Coll/TA-based materials, it is around 68 °C [107].

Collagen–tannic acid complexes have been tested as a bead material which may be potentially applied for adipose tissue regeneration by its injection after breast cancer removal. After injection, no symptoms of reactions from the immunological system were noticed. After 3 months, the implant showed incorporation into the native tissue, and fat cell growth was observed [108]. It was demonstrated that collagen–tannic acid complexes have good biocompatibility and bioactivity as they stimulate cell proliferation [109].

In scaffolds obtained from collagen cross-linked by tannic acid addition, good biocompatibility was established when 3T3 fibroblasts were used. Additionally, TA presence enhanced the healing process, which was studied on the rat animal model [110]. Collagen–tannic acid complexes have been studied as an anticancer therapy in breast cancer and melanoma treatment [111]. Bridgeman et al. reported that tannic acid presence inhibited the proliferation of A375 melanoma cells. Nevertheless, it did not function in a similar manner in non-cancerous NIH 3T3 fibroblasts. However, there is a lack of in vivo studies which would confirm the tannic acid effects on the inhibition of melanoma cancer cell growth and activity.

### 2.5. Collagen–Ferulic Acid

Ferulic acid (Figure 2) presents low cytotoxicity and has many active properties, e.g., antimicrobial, anticancer, and anti-inflammatory characteristics [112]. It has been studied for biomedical, cosmetic, and food industry applications. Ferulic acid at the concentration of 300 µL/mL showed no significant cytotoxicity. Additionally, the inhibition of platelet aggregation and an anticoagulant effect was observed [113]. Ferulic acid is a free radical scavenger and inhibits the activity of the enzymes responsible for free radical generation catalysis. It promotes angiogenesis and accelerates wound healing. However, it tends to be oxidized, which limits its industrial application. Ferulic acid may also be proposed for the improvement of biopolymer properties, since it bears hydrophilic groups which are able to interact with functional groups in a polymeric chain.

Collagen was modified by ferulic acid to obtain composite electrospun nanofibers in compositions with polycaprolactone [114]. The fabricated materials have been tested in chronic wound treatment. Ferulic acid was added as a biologically active antimicrobial agent supporting wound healing. It has been concluded that ferulic acid addition to collagen inhibits its self-association, and the inhibition rate depends on FA concentration and temperature. Ferulic acid is an inhibitor of collagen fibrillation and its propagation [115].

In comparison to tannic acid, ferulic acid is a less commonly studied phenolic acid. There is a lack of in vivo studies of materials with ferulic acid addition. Based on in vitro tests, it may be assumed that it is safe and non-toxic; however, there is a need for further biological studies of this cross-linker before it is proposed for biomedical applications.

### 2.6. Collagen–Gallic Acid

Gallic acid (Figure 3) is a phenolic acid of low molecular weight when compared to other compounds. It has been studied as a collagen modifier under the influence of which a change in the collagen secondary structure and loosening of its network was confirmed.

Studies showed that gallic acid improves bone regeneration. The healing process improvement was observed as a result of osteoprotegerin and bone morphogenetic protein-2 expression [116]. Gallic acid addition improved the physicochemical properties of collagen-based scaffolds [117]. Additionally, materials obtained from collagen modified by gallic acid are promising when discussing bone tissue regeneration. Collagen–gallic acid complexes showed a reduced degradation degree and antioxidant activity [118].

Collagen-based scaffolds cross-linked by dialdehyde chitosan in the presence of gallic acid showed high biocompatibility as they enhanced cell proliferation and adhesion properties [83]. Gallic acid was also tested as an additive for collagen cross-linked by 1-ethyl-3-(3-dimethylaminopropyl)-carbodiimide (EDC) and N-hydroxysuccinimide (NHS). Such modification inhibits collagenase activity [119].

### 2.7. Collagen with Other Phenolic Acids

Ellagic acid (Figure 4) was studied as a cross-linker for chitosan/collagen scaffolds. The obtained materials showed antioxidant activity [120]. Another polyphenol, caffeic acid (Figure 5), has a significant effect on an anti-inflammatory activity and, as a result, on wound healing [121]. Additionally, it has an antioxidant effect and stimulates collagen synthesis processes [122]. Caffeic acid prevents premature aging and shows antimicrobial activity [123]. It has been reported as an effective cross-linker in the case of chitosan/collagen hydrogels with antioxidant activity [124]. It has also been studied as a chitosan cross-linker, and the obtained films showed antioxidant and antibacterial activity [125].

### 2.8. Phenolic Acids with Other Natural Polymers

Tannic acid has been reported as an effective cross-linker for chitosan. The presence of tannic acid improves the mechanical properties of the films and decreases the degradation rate [126,127]. Moreover, TA added to chitosan improves cell viability which was determined by tests carried out, i.e., by seeding cells on a thin film surface. Tannic acid addition to chitosan results in the decrease of bacteria adhesion as well as the therapeutic release of phenolic acid into its surrounding at pH = 7.4. Thereby, such mixtures may be proposed as coatings, e.g., to cover a metal surface or act as antibacterial protection during implantation.

Agarose/tannic acid hydrogel scaffolds were fabricated for drug delivery purposes [128]. The prepared hydrogels showed anti-microbial and anti-inflammatory properties and lacked cytotoxicity. Agarose-based hydrogels with tannic acid addition also show improved mechanical properties when compared to those without TA addition. Tannic acid has also been studied in combination with starch [129], silk fibroin [130], hyaluronic acid [131], and cellulose [132]. For all the complexes, an improvement in physicochemical and biological properties was noticed.

Gallic acid–collagen complexes were reported by Bam et al. [133]. Gallic acid addition improved the physicochemical properties of collagen-based scaffolds. Gallic acid was studied as a cross-linker for chitosan, and the formed materials showed antimicrobial and antioxidant activity [134]. Gallic acid was tested as an active substance loaded in agarose/gelatin nanoparticles [135] and hyaluronic-based particles [136]. Gallic acid addition to starch was studied as beneficial to control starch product digestion [137]. It was also studied as a cross-linker for cellulose [138].

Ferulic acid was studied with collagen as an inhibiting factor in collagen fibrillation during its accumulation [139]. A ferulic acid–chitosan conjugate was fabricated, and the formed complexes showed antioxidant activity [140]. It was also studied as a modifier for chitosan/sodium alginate mixtures as an effective cross-linker [141]. This phenolic acid was also studied as a cross-linker for starch [142] and cellulose [143].

Caffeic acid has been reported as an effective cross-linker of chitosan/collagen hydrogels with antioxidant activity. It has also been studied as a chitosan cross-linker, and the obtained films showed antioxidant and antibacterial activity [144]. A caffeic acid–starch complex was studied as safe for biomedical applications [145].

Another phenolic acid, ellagic acid, was researched as a cross-linker for chitosan/collagen scaffolds. The obtained materials showed antioxidant activity [120]. Additionally, ellagic acid was used as a chitosan/sodium alginate modifier [146]. It was also added to a starch-based film as an active compound [147]. Ellagic acid–hyaluronic acid complexes were produced, and they formed biocompatible complexes [148].

Studies of phenolic acid–biopolymer complexes involve mainly the characterization of the obtained materials (films, hydrogels, scaffolds) [149,150,151,152,153,154]. However, polyphenolic acid activity in the presence of biopolymers still requires an explanation. There is a need to understand molecular interactions between a polymeric chain and phenolic acid molecules in order to propose such modifications as an alternative to the existing methods based on the use of harmful chemical factors, mainly aldehydes [155,156,157,158,159]. These have been widely studied as good cross-linkers for biopolymers; however, there is a risk of releasing toxic monomers unbound to biopolymers. They may even be released to the human body upon the degradation process. Understanding phenolic acid behavior in contact with biopolymers may lead to the development of an eco-friendly alternative for effective and safe natural polymer modifications [160,161,162,163,164].

## 3. Conclusions

Phenolic acids have been tested as collagen cross-linkers. They interact with hydrophilic groups which are present in a polymeric chain (Figure 6). As a result, complexes of collagen–phenolic acids are formed. Such complexes are used to obtain different types of materials proposed mainly for biomedical applications. The preparation of materials from collagen modified by phenolic acids has led to an improvement in the physical and chemical properties. Additionally, those materials exhibit antimicrobial and antioxidant activities. It is particularly advantageous when polyphenols are to be used as collagen cross-linkers. As effective cross-linkers, different phenolic acids have been studied, e.g., tannic, ferulic, gallic, ellagic, and caffeic acid. However, there is a lack of pre-clinical studies of collagen-based materials with phenolic acid addition. Their biological properties have been reported by in vitro tests; nevertheless, it is necessary to carry out in vivo studies to confirm their activity and determine their influence on the tissues. Based on the available scientific reports, we would like to make a claim that phenolic acids are promising collagen modifiers.

## Figures and Tables

**Figure 1 materials-13-03641-f001:**
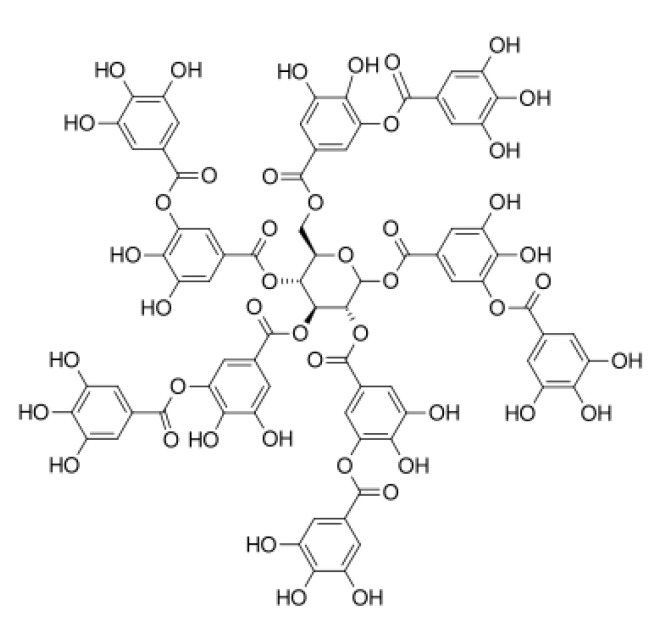
The chemical structure of tannic acid.

**Figure 2 materials-13-03641-f002:**
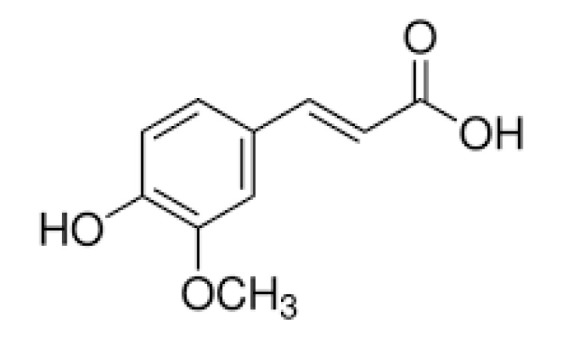
The chemical structure of ferulic acid.

**Figure 3 materials-13-03641-f003:**
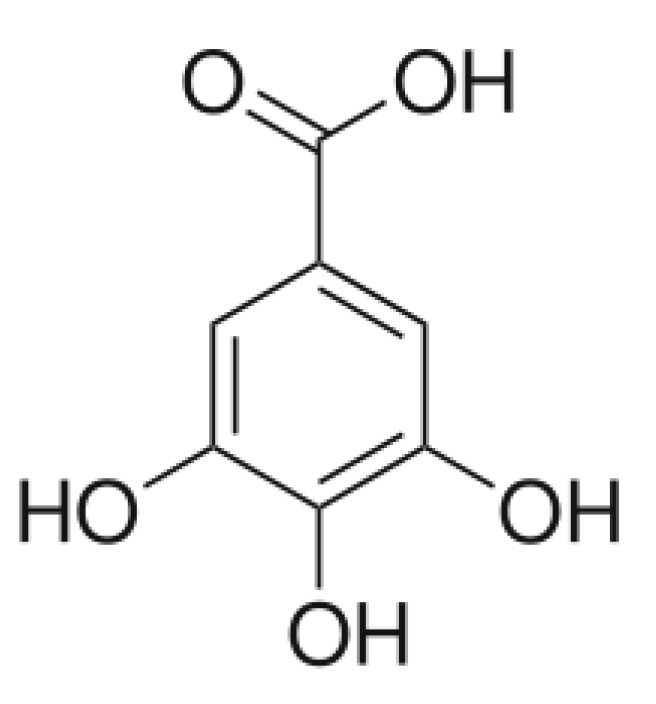
The chemical structure of gallic acid.

**Figure 4 materials-13-03641-f004:**
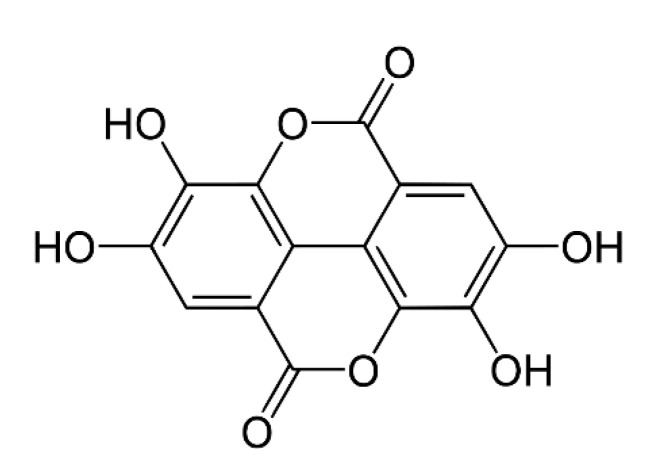
The chemical structure of ellagic acid.

**Figure 5 materials-13-03641-f005:**
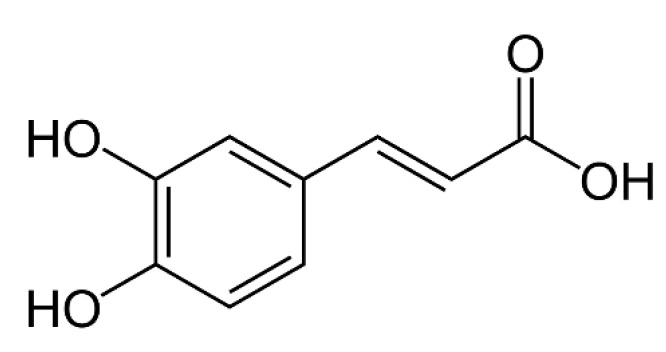
The chemical structure of caffeic acid.

**Figure 6 materials-13-03641-f006:**
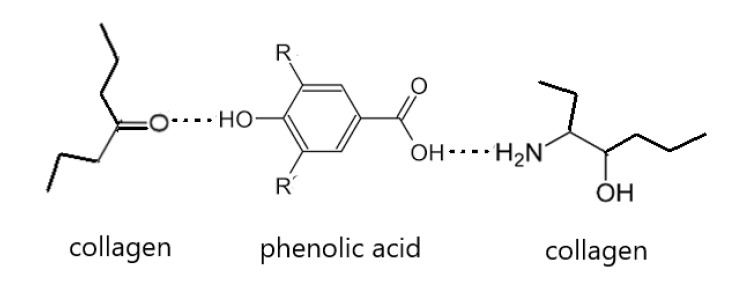
The schematic interaction of collagen with phenolic acids.

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
