# Peer review of "Collagen-Based Materials Modified by Phenolic Acids—A Review"

_materials, 2020, doi:10.3390/ma13163641_

Round 1

Reviewer 1 Report

After English language are corrected, I recommend that the MS will be accepted for publication in Materials.

Author Response

Dear Reviewer, 

We would like to thank for the comments to our manuscript submitted to Materials journal for review process. We would like to thank also the Editor that gave me chance to correct our manuscript. All changes made in our manuscript are written in red. Please find below our answer to the comments.

Reviewer #1: (round 1)

After English language are corrected, I recommend that the MS will be accepted for publication in Materials

Thank you very much. Language is now improved.

Reviewer 2 Report

The goal of the manuscript entitled “Collagen-based materials modified by phenolic acids – A review” by Kaczmarek & Mazur is to present an exhaustive description for different biomaterials based on collagen and phenolic acid. The manuscript is submitted as a review type article, but in my opinion, it needs to be significantly improved in order to be suitable for the Materials journal.

  1. The results obtained in the field presented in the first part of the text (Introduction section) are described in a “narrator description mode”, very summary, without their systematic description (short presentation of the results taken from other articles) and without the support of bibliographic references.
  2. Bibliographic references were not collected properly in order to incorporate the existing results, problems and challenges of the targeted research field. The first bibliographic reference is mentioned only at second part of page 3. They are not well spread in the text (in sections 2.1-2.5 there are not mentioned any relevant bibliographic references), and there are places where 10-11 references are included in a single bracket.
  3. The presentation of the collagen and its related materials are started to be presented only in page 5, in section 2.6, although the aim of the author is to present the properties of collagen-based materials.
  4. The whole structure of the manuscript does not fit with a usual, review type article form, where, first, are presented the most relevant result from the field, followed by the description of the problems which are still difficult to be solved as well as challenges that should be comprehensively addressed. Finally, some important perspectives also need to be formulated.

I consider that the present manuscript needs to be completely reconsidered in order to have the form of a review type article.  

The present manuscript might be suitable for publishing in the Materials journal only after comprehensive revision and a new submission.

Author Response

Dear Reviewer, 

We would like to thank for the comments to our manuscript submitted to Materials journal for review process. We would like to thank also the Editor that gave me chance to correct our manuscript. All changes made in our manuscript are written in red. Please find below our answer to the comments.

Reviewer #2: (round 1)

The results obtained in the field presented in the first part of the text (Introduction section) are described in a “narrator description mode”, very summary, without their systematic description (short presentation of the results taken from other articles) and without the support of bibliographic references.

Thank you for the suggestion. The references are now added.

Bibliographic references were not collected properly in order to incorporate the existing results, problems and challenges of the targeted research field. The first bibliographic reference is mentioned only at second part of page 3. They are not well spread in the text (in sections 2.1-2.5 there are not mentioned any relevant bibliographic references), and there are places where 10-11 references are included in a single bracket.

References are now corrected.

The presentation of the collagen and its related materials are started to be presented only in page 5, in section 2.6, although the aim of the author is to present the properties of collagen-based materials.

I appreciate your comments. It has been now changed.

The whole structure of the manuscript does not fit with a usual, review type article form, where, first, are presented the most relevant result from the field, followed by the description of the problems which are still difficult to be solved as well as challenges that should be comprehensively addressed. Finally, some important perspectives also need to be formulated.

We appreciate your comment. Manuscript is now improves based on the reviewers’ comments. We hope that present form is acceptable.

Reviewer 3 Report

Major revision(see attached file)

Author Response

Dear Reviewer, 

We would like to thank for the comments to our manuscript submitted to Materials journal for review process. We would like to thank also the Editor that gave me chance to correct our manuscript. All changes made in our manuscript are written in red. Please find below our answer to the comments.

Reviewer #3: (round 1)

Critical remarks:

- Abstract

The sentence: “Collagen-based materials constitute one of the most widely studied types of materials for biomedical applications” is confusing due to repetitive “materials”. Change to biomaterials.

Thank you for the suggestion. It is now corrected.

- Introduction: There is completely lack on references. For example after such sentence as:

“The first bone marrow transplant in leukemia in 1956 is considered the beginning of regenerative medicine.” Reference must be inserted, as well as in other similar cases through the text. For example, here “Various methods have been already tested as the 115 complexation by metal (e.g. Fe(III), Pb(II), Cu(II)) or the use of cross-linkers.”

Thank you for the suggestions. It is now corrected.

- The text between Line 22 and 127 must be shortened up to 50%.

Manuscript has been now modified. We hope the present form is acceptable.

- - You review is regarding to Collagen and not to chitosan and other biopolymers you have discussed. Thus, remove subchapters concerning chitosan, cellulose, etc.

We appreciate the suggestion. Subchapters about other biopolymers are now removed.

- Line 284: collagen has been reported to be present in fungi. Read Ehrlich, 2010.

- Line 285: Why you wrote “and even from sea sponges”? The first collagen in multicellular organisms has been found in marine (not sea) sponges. See: Ehrlich et al (2010) Mineralization of the meter–long biosilica structures of glass sponges is template on hydroxylated collagen. Nature Chemistry 2:1084–1088. There is lack on recent papers concerning jellyfish collagens too.

Reference list is now improved.

- Figs 1,2 and 3 are not necessary, principally.

Figures are removed in present form of paper.

- Subchapter 2.6.2. There is lack on references again.

References are now added.

- Lines 361-374 – lack of references, again

It is now corrected.

- The title “Collagen with other phenolic acids” must be changed

Based on the reviews, we decided to remove paragraphs about chitosan, hyaluronic acid etc to underline that the review is about collagen. Thereby, we will leave the title and we hope that now it is acceptable.

- It is strongly recommended prior to rewrite your review to read and to discuss following review papers on collagens: Kadler et al (2007) J Cell Sci, 120, 1955 ;Ehrlich H. (2010) Chitin and collagen as universal and alternative templates in biomineralization. International Geology Review 52:661–699; Ehrlich et al(2018) Collagens of poriferan origin. Marine Drugs 16:79 ; Jesionowski et al(2018) Marine spongin: naturally prefabricated 3D scaffold–based biomaterial. Marine Drugs 16:88 , as well as https://doi.org/10.1080/10408398.2020.1751585

We appreciate your recommendations. Mentioned articles have been now added.

Also, there is lack on references from the papers of Koob and Trotter (1989-1996) concerning the discovery of tanned collagen in marine invertebrates. See as example: Koob, T.J., Cox, D.L. Stabilization and sclerotization ofRaja erinacea egg capsule proteins. Environ Biol Fish 38, 151–157 (1993). https://doi.org/10.1007/BF00842911. Additionally, I recommend to read:

https://doi.org/10.1016/0305-0491(76)90214-5; DOI: 10.1016/0022-0981(87)90124-9; DOI: 10.1016/0040-8166(92)90048-c ; doi: 10.1006/bbrc.1998.9282; doi:10.1016/j.ijbiomac.2007.01.002;

DOI: 10.1100/tsw.2003.13; DOI: 10.1016/j.ijbiomac.2010.08.003; https://doi.org/10.1007/s13770-014-0075-y

We would like to thank you for recommendation of such interesting papers. They have been now citated.

- You wrote that collagen possess low thermal stability. However, it is necessary to mention in your review that collagen of demosponge origin is thermostable up to 360°C. References:

Szatkowski et al (2018) Extreme biomimetics: carbonized 3D spongin scaffold as a novel support for nanostructured manganese oxide (IV) and its electrochemical applications. Nano Research 11(8):4199–4214; Petrenko et al (2019) Extreme Biomimetics: Preservation of molecular detail in centimetre scale samples of biological meshes laid down by sponges. Science Advances, 5:eaax2805

Thank you very much for this important issue. It is now mentioned in the review.

Recommended references on practical application of collagens:

Heinemann et al (2007a) Ultrastructural studies on the collagen of the marine sponge Chondrosia reniformis Nardo. Biomacromolecules 8:3452–3457

Heinemann et al (2007b) Biomimetically inspired hybrid materials based on silicified collagen. International Journal of Materials Research 98:603–608

Heinemann et al. (2007c) A novel biomimetic hybrid material made of silicified collagen: perspectives for bone replacement. Advanced Engineering Materials 9(12):1061–1068

Ehrlich et al (2008) Nanostructural organisation of naturally occurring composites: Part I. Silica–collagen–based biocomposites. Journal of Nanomaterials 2008, Article ID 623838, 8 pages, doi: 10.1155/2008/670235)

Ehrlich et al (2009) Modification of collagen in vitro with respect to N–carboxymethyllysine. International Journal of Biological Macromolecules 44:51–56

Ehrlich et al (2010) Carboxymethylation of collagen with respect to Ca–phosphate phases formation. Journal of Biomedical Materials Research, Part B. Applied Materials 92(2):542–551

Szatkowski et al (2015) Novel nanostructured hematite–spongin composite developed using extreme biomimetic approach. RSC Advances 5:79031–79040

Tsurkan, et al (2020) Modern scaffolding strategies based on naturally pre-fabricated 3D biomaterials of poriferan origin. Applied Physics A (2020)

We appreciate your effort. We decided to added selected papers in the reference list.

Reviewer 4 Report

1- The abstract is not clear authors need to put more effort to summarize their work more concisely.

2- Author should focus on introduction, at present there are no correlation between paragraphs and the rationale of the work performed. The non-expert on the topic will have a difficult time following this introduction and understanding its relevance to this paper.

3- Author should include recent references in the text, at present very poorly references are cited.

4- I am surprised why Author mentioned about Chitosan, hyaluronic acid, cellulose, alginate and Starch.

5- It will be good if author could explain the chemical interaction of collagen with phenolic acids and how it enhance therapeutic potential.

6- Discussion and conclusions need to rewrite.

7- Future perspective will be nice addition.

Author Response

Dear Reviewer, 

We would like to thank for the comments to our manuscript submitted to Materials journal for review process. We would like to thank also the Editor that gave me chance to correct our manuscript. All changes made in our manuscript are written in red. Please find below our answer to the comments.

Reviewer #4: (round 1)

1- The abstract is not clear authors need to put more effort to summarize their work more concisely.

Manuscript is now modified. We hope its is acceptable.

2- Author should focus on introduction, at present there are no correlation between paragraphs and the rationale of the work performed. The non-expert on the topic will have a difficult time following this introduction and understanding its relevance to this paper.

Thank you for the comment. It has been now modified.

3- Author should include recent references in the text, at present very poorly references are cited.

Thank you for the suggestion. It is now improved.

4- I am surprised why Author mentioned about Chitosan, hyaluronic acid, cellulose, alginate and Starch.

Dear reviewer, this section is now removed.

5- It will be good if author could explain the chemical interaction of collagen with phenolic acids and how it enhance therapeutic potential.

Complex of collagen-phenolic acid is formed by strong hydrogen interactions. The main issue is that such complexes possess novel properties compared to pure collagen as antibacterial, antiviral etc. Also, the complex formation improves collagen physicochemical properties. All of these information are now present in the paper.

6- Discussion and conclusions need to rewrite.

Manuscript has been now modified.

7- Future perspective will be nice addition.

In our opinion there is a need to carry out the in vivo studies of collagen-based complexes with phenolic acid. Mainly, researchers are focused on the primary studies with the use of different cell lines. In our opinion, important is to study the influence of phenolic acid implantation on the tissues.

Round 2

Reviewer 2 Report

This new revised version of the manuscript shows significant improvements compared to the first submitted version of the manuscript. The authors have given satisfactory solutions and explanations for the addressed questions related to their original work.

The manuscript can be considered for publishing in Materials journal in the present form.

Author Response

Thank you very much.

Reviewer 3 Report

Accept

Author Response

Thank you very much.

Reviewer 4 Report

Dear Authors,

Happy to see the changes you have incorporated, it will be nice if you could give one figure for the chemical interaction of collagen with phenolic acids and how it enhance therapeutic potential.

Author Response

Dear Reviewer, 

We would like to thank for the comments to our manuscript submitted to Materials journal for review process. We would like to thank also the Editor that gave me chance to correct our manuscript. All changes made in our manuscript are written in yellow. Please find below our answer to the comments.

Reviewer #4: (round 2)

Happy to see the changes you have incorporated, it will be nice if you could give one figure for the chemical interaction of collagen with phenolic acids and how it enhance therapeutic potential.

Thank you for the suggestion. In the manuscript you can find now Figure 6. The therapeutic potential is described in the paper where the advantages of phenolic acid addition to the collagen in written. We hope that it is acceptable.